# Aortopexy for Tracheomalacia in Children: A Systematic Review and Meta-Analysis

**DOI:** 10.3390/jcm14041367

**Published:** 2025-02-18

**Authors:** Andrea Zanini, Stefano Mazzoleni, Luigi Arcieri, Raffaele Giordano, Stefano Marianeschi, Francesco Macchini

**Affiliations:** 1Pediatric Surgery, ASST Grande Ospedale Metropolitano Niguarda, 20162 Milan, Italy; andrea.zanini@ospedaleniguarda.it (A.Z.);; 2Pediatric Cardiac Surgery, ASST Grande Ospedale Metropolitano Niguarda, 20162 Milan, Italy; 3Advanced Biomedical Sciences, Adult and Pediatric Cardiac Surgery, University of Naples Federico II, Via Pansini 5, 80131 Naples, Italy

**Keywords:** aortopexy, tracheomalacia, children, pediatric cardiac csurgery

## Abstract

**Introduction:** Aortopexy is the most common surgical option for the treatment of severe tracheomalacia (TM) in children. Despite being described over 75 years ago, several aspects of the procedure remain controversial. **Materials and Methods:** A systematic review of aortopexy was conducted following the PRISMA guidelines. All original articles describing at least one case of aortopexy performed in children for the treatment of TM were included. Patients’ characteristics, surgical indications, approaches and details were collected. Outcomes were reviewed, and potential factors associated with procedural success were investigated. **Results:** Of 243 papers, 17 were included in this review, encompassing a total of 473 patients who underwent aortopexy. Of these, 65.3% were male, with a mean age at surgery of 12.2 months (5 days–18 years). Primary TM accounted for 7.9% of cases, while 92.1% were secondary to other anomalies, including esophageal atresia (54.1%), aberrant innominate artery (16.1%) and vascular rings (8.3%). There is a general consensus on the surgical indication for TM with severe symptoms, although the definition of severity is not well established. Overall success was 84%, but 40.8% of patients experienced some persistent symptoms. Sternotomy and thoracotomy were the most successful approaches (92.6% and 84.2%, respectively). Thoracoscopy had a similar success rate to sternotomy when pericardiotomy, thymectomy and pledgeted suture were performed (86.4% vs. 92.6% *p* = 0.41). **Conclusions:** Aortopexy is associated with good outcomes, but no evidence-based guidelines are currently available. Success appears to be associated with specific surgical steps rather than the surgical approach. Prospective studies are desirable for establishing precise guidelines.

## 1. Introduction

Aortopexy, first described by Gross in 1948 [1], remains the most commonly used surgical option for the treatment of tracheomalacia (TM) [2]. Although aortopexy is widely performed, many aspects of the procedure are still controversial [3].

Indications for surgical treatment of tracheomalacia are not uniform [4]. Surgery is usually reserved for the more severe form of TM who present with severe symptoms [2]. However, a standardized definition of the symptoms that should prompt considering a surgical option is currently lacking [3]. A consensus on the threshold of TM severity that indicates the surgical correction is not present either. Moreover, the classification of TM severity itself is not standardized [5,6].

Furthermore, while aortopexy was traditionally performed via median sternotomy, various approaches such as partial sternotomy, cervical and, more recently, thoracoscopic techniques have been reported [7]. Currently, no superiority of one surgical approach has been proven in terms of success rate. In addition to the surgical approach, some technical details also vary across published experiences: whether a total thymectomy is performed, the use of pericardiotomy and the number and location of sutures [8,9,10].

Finally, the majority of the published literature on aortopexy is derived from retrospective single-center experiences [7].

In an attempt to address these controversies, we conducted a systematic review of the past 10 years. This review aims to evaluate the indication for aortopexy for the treatment of TM, to provide a comprehensive overview of the outcomes and to identify potential factors associated with procedural success. The authors suggest that a systematic review of the available current experiences could provide more reliable data for clinical decision-making and contribute to the development of a standardized approach for treating pediatric tracheomalacia.

## 2. Materials and Methods

### 2.1. Study Design

We conducted a systematic review following the PRISMA guidelines [11] of all original articles published in the past 10 years (2014–2024), reporting at least one case of aortopexy for the treatment of tracheomalacia in the pediatric age group (0–18 years).

### 2.2. Search Strategy

A search on PubMed was performed using the following query:

(aortopexy) AND ((pediatric[MeSH Terms]) OR (infant[MeSH Terms]) OR (children[MeSH Terms])).

An additional search using the only keyword “aortopexy” was conducted to identify any possible additional study. We did not use any filter and limit to keep the search as wide as possible, thus minimizing the possibility of missing relevant publications.

The date of the last search was 20 December 2024.

Summarizing, these are the eligibility criteria for paper selection:

Original report;

At least one reported case of aortopexy;

Paper published between 2014 and 2024.

Every study that met the eligibility criteria was reviewed by two authors (A.Z. and S.Maz.) for the final inclusion in the present review.

Exclusion criteria were not English papers, studies on adults, indications other than tracheomalacia and papers lacking sufficient details.

Overlapping papers were excluded, including only the most recent publication from the same center. Papers excluded for this reason were nonetheless reviewed to retrieve any useful data that might be missing in the most recent report.

A risk of bias assessment was conducted for all 17 included studies using the ROBINS-I (Risk of Bias in Non-randomized Studies of Interventions) tool [6]. Two authors appraised all studies (A.Z. and S.Maz). Inter-rater agreement was calculated as the percentage of the agreed opinion on the domain level. Discrepancies were discussed with a third author (F.M.).

Results are visually reported using the robvis V2 tool [12].

All included papers were independently and individually reviewed by two authors (A.Z. and S.Maz.) for the data extraction. Data were collected on an Excel database that was prepared beforehand with the desired information to be extracted.

After the data extraction, the two databases were compared; any differences between the two versions were addressed by re-reviewing the relevant paper together with a third author (L.A.).

### 2.3. Data Collection

Patients’ characteristics and demographic data were reviewed and collected.

Surgical details were collected: surgical approach (sternotomy, thoracotomy, cervicotomy, thoracoscopy); whether thymectomy was performed (and if so, whether total or partial); pericardiotomy, number, type and site of sutures placed; and the use of intra-operative tracheo-bronchoscopy. Moreover, any additional procedures, such as anterior tracheopexy, were noted.

Surgical complications were collected. Success, failure and mortality were recorded. Success was defined as improvement of the presenting symptoms. When reported, success was divided into total and partial. Total success was defined as the complete resolution of symptoms. Partial success was defined as improved but present symptoms.

When not specified, failure was defined as the need for re-operation or mortality related to tracheomalacia. The length of follow-up was collected when reported by the authors.

When possible, the success rate was assessed according to the surgical approach and the techniques to identify any potential predictive factors.

In case of missing data, the descriptive analysis was performed on the available data. The amount of missing data and the number of papers upon which statistics were performed are mentioned in the Results section.

Statistical analysis was performed using Microsoft Excel. Descriptive statistics was used for demographic data, *t*-test for average comparison, ANOVA for multiple group comparison and Fisher’s exact test for qualitative variables as sample size and several categories were expected to be small [13].

## 3. Results

### 3.1. Literature Search Results

Out of the 243 analyzed papers, 17 were included in the present review [9,10,14,15,16,17,18,19,20,21,22,23,24,25,26,27,28,29]. The paper selection pathway is reported in Figure 1. Not all papers reported all the data we planned to analyze.

A summary of the papers and main findings included in this review is reported in Table 1.

Figure 2 and Figure 3 summarize the risk of bias in the included studies. Most studies were retrospective case series or cohorts, inherently at higher risk of bias. The inter-rater agreement was 74% (88/119) and achieved full inter-rater agreement after discussion of discrepancies with the third author. Overall, 14 studies (82%) had a serious risk of bias, while 3 (18%) had a moderate risk. The primary sources of bias included the following:Confounding Bias—The lack of randomization and retrospective designs limited control over patient characteristics and surgical evolution. Only three studies applied statistical adjustments;Classification Bias—Surgical approaches were well defined, but differences in surgeon experience and adjunctive procedures may have influenced outcomes;Selection Bias—Inclusion criteria varied, and long study periods introduced time-related selection bias. Some studies excluded lost-to-follow-up patients without adequate discussion of their characteristics, potentially introducing bias;Deviation from Intended Intervention Bias—Reporting on deviations (e.g., need for tracheostomy, revision surgeries) was inconsistent, and post-operative management varied across institutions;Missing Data Bias—Missing data was a significant concern, particularly in long-term follow-up assessments. While a few studies reported comprehensive follow-up data, many had high attrition rates, with follow-up periods ranging from as little as 6 weeks to over 14 years;Outcome Measurement Bias—Assessments relied on subjective symptom resolution, often without standardized scoring or objective follow-up (e.g., bronchoscopy). Parent-reported outcomes may introduce recall bias;Reporting Bias—Selective reporting favored positive findings, and few studies had pre-registered analysis plans, increasing the risk of post hoc analyses

### 3.2. Patients’ Epidemiology

A total of 473 patients’ data were collected. Within the specified patient gender, 64.1% were male (270/421) and 35.9% female (151/421). The mean age at surgery was 12.2 months, ranging from 5 days to 18 years. Weight at surgery was seldom reported and therefore not included in the review.

Out of the 300 patients in whom the site of malacia was specified, 65.3% had pure tracheomalacia (196/300), 29.0% had tracheo-bronchomalacia (87/300) and 5.7% had bronchomalacia (17/300).

Out of 417 patients, 7.9% (33/417) had primary TM, while 92.1% (384/417) had TM secondary to other anomalies. Among the secondary TM cases, 54.2% (208/384) were associated with esophageal atresia, 16.1% (62/384) were due to innominate artery compression, 8.3% (32/384) were due to vascular rings and in 19.8%, the etiology was not specified.

### 3.3. Symptoms

Presenting symptoms were reported in 241 patients. The most common symptoms were ALTE (apparent life-threatening event)/acute respiratory failure in 64.7% (156/241), followed by dyspnea in 27.8% (67/241), failure to extubate in 15.4% (37/241) and recurrent infections in 9.5% of patients.

Eight of the 17 papers reported surgical indications, which were severe tracheomalacia with severe symptoms. However, neither the severity of TM nor the symptoms were defined. All but one diagnosed TM using pre-operative bronchoscopy, while the other relied on CT.

### 3.4. Surgical Approach

The surgical approach was mentioned in 453 cases. Thoracoscopy accounted for 10.2% of cases (46/453), while 89.8% of cases (407/453) were performed with open techniques: sternotomy (either partial or total) was used in 63.1% of cases (286/453), antero-lateral thoracotomy in 16.3% (74/453) and cervicotomy in 7.1% (32/453); a total of 15 cases used open techniques not further specified.

Thymectomy was not mentioned in four papers. Of the 13 authors who specified, 8 always performed thymectomy (61.5%), 3 sometimes performed it (23.1%) and 2 did not perform it (15.4%). Of the 11 papers where thymectomy was performed, 5 reported either partial or total thymectomy, 3 reported partial thymectomy and 3 reported total thymectomy. As a result, thymectomy was performed in 88.8% of the specified cases (230/259), but it was not possible to assess the exact number of patients undergoing a partial or total thymectomy. Thymectomy was performed in 47.8% (22/46) of MIS cases and in 97.7% (208/213) of open cases.

The pericardium was always opened by three authors, left intact by two authors and sometimes opened by two authors. Ten papers did not specify whether the pericardiotomy was performed. As a result, pericardiotomy was performed in 224 patients (92.2%) and left intact in 19 (7.8%).

Pericardiotomy was performed in 58.7% (27/46) of MIS cases and left intact in 41.3% (19/46). On the contrary, the pericardium was always opened in open approaches (197/197).

The number of traction sutures ranged from two to seven, with most authors placing either two or three. Sutures were always non-absorbable, either Prolene^®^ or Ethibond^®^. Nine authors used pledgeted sutures, while two used simple sutures. As a result, 93.4% (338/362) of patients had pledgeted sutures, while 6.6% (24/362) had simple sutures.

Pledgeted sutures were used in 47.8% (22/46) of MIS cases and always used in open cases (316/316).

The innominate artery was always suspended by two authors and sometimes by four, while six authors did not perform IA suspension.

Concomitant procedures included anterior or posterior tracheopexy, ring division and pulmonary artery suspension.

### 3.5. Aortopexy Results

The overall success rate of aortopexy was 84.0%, with 236 patients reported to have partial or total symptom resolutions and 45 reported as unsuccessful. Out of the 174 specified successful cases, 59.2% (102/174) had total relief, while 40.8% (71/174) had partial symptom relief.

The mortality rate was 6.6% (31 cases), and the reported complication rate was 16.6%. Risk factors for mortality and complications were not analyzed due to a lack of data.

The success rate for MIS was 73.9%, significantly lower than for open techniques (86%, *p* = 0.041). The success rate was significantly related to surgical approach: MIS 73.9%, sternotomy 92.6%, thoracotomy 84.2% and cervicotomy 68.8% (*p* = 0.025). There was no significant difference between sternotomy and thoracotomy. Among open approaches, cervicotomy has a significantly lower success rate compared to other open techniques (*p* = 0.001).

Variables like pericardiotomy, thymectomy and pledgeted sutures were not consistent only within the MIS population. Out of the 46 cases, 22 underwent pericardiotomy, thymectomy and pledgeted sutures, while 24 had their pericardium left intact, thymus not removed and simple suture placed. The success rate between these two populations was 86.4% and 62.5%, respectively (*p* = 0.06). The success rate of the MIS cases following the standard open principles did not significantly differ from the success rate of sternotomy (86.4% vs. 92.6%; *p* > 0.05).

No other independent predictor factors have been identified.

## 4. Discussion

Despite being first described over 75 years ago and having been reported in hundreds of patients [7], aortopexy for the treatment of tracheomalacia remains associated with several controversies regarding almost every aspect of the surgical procedure, from indications to surgical technical details [2].

A Cochrane review was published in 2005 aiming to evaluate the efficacy of medical and surgical therapies for children with intrinsic tracheomalacia. No RCTs were found, and the authors concluded the absence of evidence to support any of the therapies used for tracheomalacia [30]. A subsequent update published in 2012 included an RCT on the use of nebulized recombinant deoxyribonuclease (rhDNase) in 40 children with airway malacia. The authors stated that this therapy could not be recommended given the cost and likely harmful effect. They concluded that the decision for either medical or surgical therapy would have to be made on an individual basis [31]. The authors also expressed concern that any RCT on surgically based management would ever be available for severe life-threatening tracheomalacia while RCTs on less severe diseases would be needed.

Since 2012, no RCT on surgical management of tracheomalacia has indeed been published.

The above-mentioned Cochrane reviews, however, specifically focused on the treatment of intrinsic (primary) tracheomalacia, excluding from the scope of their review the treatment of the secondary tracheomalacias, which accounts for the vast majority of the disease [31].

Nonetheless, the criteria to distinguish primary and secondary TM differ across the available literature. Some authors indeed classify the “innominate artery (IA) compression” as a form of secondary tracheomalacia, while others believe that the IA compression is rather just a radiological appearance of an intrinsic airway malacia [32].

More recently, a statement on tracheomalacia and bronchomalacia in children from the European Respiratory Society was published in 2019. The authors concluded that there was still poor evidence to provide clear guidelines on the treatment for tracheomalacia [3,31]. Surgical correction was considered an option for the treatment of tracheomalacia associated with severe symptoms [3]. This is consistent with our findings: according to our review, the majority of authors agreed on the need for surgical correction in patients presenting with life-threatening symptoms or even a single episode of a “dying spell” [2]. However, all the papers in this review were retrospective in nature, and when specified, the authors simply reported the individual indications for surgery for their series, rarely stating whether there were pre-existent institutional standardized indications for surgery.

While there seems to be a general consensus on the need for surgical correction for more severe cases, the indication for surgery in patients with non-life-threatening symptoms is less clear [3]. Moreover, although the majority of authors offered aortopexy to treat TM when symptoms were “severe”, the definition of such severity was not well defined for symptoms like “recurrent infections”, “respiratory distress” and “stridor”. A definition was proposed by Fraga et al., who advised surgery in patients with more than three episodes of pneumonia per year, but this definition is not universally shared [2]. To date, there is no univocal consensus on the symptoms constellation that should prompt the decision for surgical correction of tracheomalacia [33].

Most of the aortopexies performed for non-life-threatening TM were indicated based on an individual basis. In our experience, a multidisciplinary individual case discussion together with the family is paramount for the decision of surgical correction of a tracheomalacia.

Furthermore, a detailed diagnostic work-up is crucial prior to considering surgical correction of tracheomalacia in order to exclude other possible conditions that may explain the symptoms and to plan the best surgical option [2,3]. Particularly in the TM associated with esophageal atresia, it is imperative to rule out gastroesophageal reflux and recurrent tracheo-esophageal fistula before attributing the symptoms to TM [34]. Once GER is ruled out, a bronchoscopy is useful for both the diagnosis of recurrent TEF or TM. Endoscopic evaluation of the tracheal collapse is indeed the most common way of diagnosing airway malacia and classifying its severity [6].

Although most authors in our review offer aortopexy for symptomatic severe tracheomalacia, the definition of TM severity itself is not always specified. Among the authors who define TM severity, there is consensus on classifying TM severity based on rigid or flexible bronchoscopy by the degree of tracheal collapse [3]. The majority adhere to the tracheal lumen reduction thresholds reported by the ERS statement: normal <50%, mild 50–75%, moderate TM 75–90 and severe TM >90% [3]. However, these thresholds are not universally shared [5,6,7,8,9,10,11,12,13,14,15,16,17,18,19,20,21,22].

In any case, there are two major pitfalls with the endoscopic classification of tracheomalacia. Firstly, the percentage of lumen reduction is a subjective assessment. It has been shown that different physicians may estimate different percentages of lumen collapse on the same endoscopy, making this method not entirely reliable [6,35]. This inter-subject variability may also contribute to the difficulty in comparing outcomes and findings among different centers. Furthermore, some authors claim that rigid bronchoscopy may underestimate the tracheal collapse as the result of the stenting provided by the rigid bronchoscope, making comparison even less reliable [5].

Another diagnostic tool used to diagnose and classify tracheomalacia is CT. Whether CT has the clear advantage of being more objective and comparable, it has the downside of slightly underestimating the degree of tracheal collapse, thus carrying the risk of missing some TM cases, especially if a dynamic CT is not obtained [3].

Another pitfall common to both radiological and endoscopic TM classification is that the anatomical severity does not always correspond to the clinical impact [3]. This is corroborated by the findings of our review, which showed that even patients with moderate TM severity have been reported to undergo aortopexy for severe symptoms [20]. Therefore, indication for surgery needs to be based on the combination of either endoscopic or radiological evidence of TM in association with severe symptoms.

Although medical therapy was not one of the aims of our review, it is interesting that the authors did not report any medical treatment trials before surgery.

Antibiotics are a cornerstone in the treatment of TM-induced infections [36]. The tracheal collapse causes recurrent vibration of the membranous trachea, often recognizable from the classic “barking cough”. This is believed to lead to irritation of the airway, reduced clearance and, ultimately, infection. Ciliary clearance may also be impaired by some degree of squamous metaplasia, which has been shown to develop over time in TM patients [3].

Although effective in treating the majority of TM-related infections, antibiotics alone are associated with a high recurrence rate, as they do not treat the underlying predisposing factor [31,36].

Medical therapies that help soften secretions, such as ipratropium bromide and DNase have been used with the theoretical assumption that improving airway clearance might reduce respiratory infections. However, not enough evidence is available to support their routine usage [3,31].

Ventilatory pressure support may have a role in TM as they have a pathophysiological rationale for preventing the airway from collapsing. CPAP is the most commonly used method [37]. Although CPAP seems to be associated with good results while in use, it is not possible to maintain a child on continuous CPAP without a tracheostomy. Moreover, CPAP is not always well tolerated by children and, therefore, its use is limited [3].

Conservative options, therefore, seem to apply to mild or moderate cases of TM, while surgery is preferred for severe cases [3].

Our review’s population characteristics confirm that TM is mostly secondary to other conditions, EA/TOF being the most common. Tracheomalacia has been reported to affect between 11 and 33% of EA cases, being more commonly associated with those EA cases with TOF [38]. However, the true incidence may be underestimated [39].

The pathophysiological reasons for this association are controversial. Some authors postulate an intrinsic defect in the tracheal development as a result of some common impairment in the separation between the embryonic trachea and esophagus. Others suggest that the pre-atretic dilated esophageal stump may compress the trachea. This may alter the C shape of the cartilaginous rings into a more open “D” or “U” shape, thus modifying the ring-to-membrane ratio. This hypothesis is supported by the observation that the malacia is often located in the tracheal segment proximal to the original tracheo-esophageal fistula rather than in proximity to it.

Some others have postulated that TM may occur as the result of some insult during the surgical repair of esophageal atresia in the neonatal period [40,41].

In a recent study by van Hal et al., the sensitivity of pre-operative bronchoscopy for TM diagnosis was only 50%, thus supporting the hypothesis that surgical correction of EA may induce or increase the severity of TM [40]. Moreover, in their series, more than half of patients born with esophageal atresia were diagnosed with tracheomalacia within the first year of life. Thus, the findings from our review, in our opinion, should prompt the extension of the endoscopic surveillance in the EA/TOF follow-up to the respiratory tract and not just the upper gastro-intestinal tract [42,43].

Other commonly associated abnormalities are cardiovascular, with tetralogy of Fallot and vascular rings being the most frequent. In these cases, the pathophysiological hypothesis is of direct extrinsic tracheal compression by the aberrant great vessels [2].

The natural history of tracheomalacia is also controversial. While TM has traditionally been considered to improve over time, unfortunately, not all children seem to outgrow TM [3,6]. The rationale behind the improvement with growth is that a bigger tracheal diameter and a more rigid ring will prevent the pars membranacea protrusion, guaranteeing good air flow and mucous ciliary clearance [3]. However, lately, this assumption has been challenged as not all patients showed symptom improvement over time [33]. In our review, the age at surgery was an average of 10 months, although it ranged widely from 5 days to 18 years. We believe that this corroborates the fact that patients do not always outgrow tracheomalacia, as previously believed [44]. However, no prospective studies are available to provide predictors of TM evolution.

Aortopexy is performed openly in the vast majority of cases, with only 10% of patients reported to have undergone thoracoscopic repair. Among open approaches, sternotomy and thoracotomy are the most commonly performed. In open approaches, almost every patient undergoes thymectomy, pericardiotomy and placement of pledgeted suture, while only Sutton et al. always followed these steps in the thoracoscopic approach [8].

Suspension of the innominate artery is not always performed. Although it may be due to different TM locations, some authors believe that innominate artery suspension should be considered a key step of the surgery [9]. Unfortunately, not enough data were available in the current review to investigate this factor.

Aortopexy has a good overall success rate with relatively low mortality and complication rates. Although there were not enough data to perform statistical analysis, the majority of mortality occurred in patients with comorbidities, mostly cardiac in nature.

Interestingly, despite most patients experiencing clinical improvement, over 40% remained partially symptomatic. In our opinion, this reflects the complexity of the physiopathology correlated to tracheomalacia, which goes beyond pure mechanical tracheal obstruction [3]. Aortopexy maintains the trachea open but does not directly address the tracheal wall. Other surgical options have been proposed and seem associated with good results, although no comparative studies are available to determine the superiority of one operation [44].

Our review showed that the success rate of aortopexy is significantly dependent on the surgical approach, with the open approach being more successful than thoracoscopy overall. However, when examining in detail, the superiority of the open approach concerns sternotomy and thoracotomy only, while cervicotomy has the lowest success rate. This may be due to the difficulty in reaching a proximal enough location for the suspension of the compressing vessel.

Several reasons have been postulated to explain the lower success rate of the thoracoscopic approach [9,10]. The learning curve may play a role, as suggested by the excellent results recently reported by Sutton et al., who documented the largest series of thoracoscopic aortopexy to date [27]. However, technical factors may also contribute to the reported inferiority of the thoracoscopic approach, such as not always opening the pericardium and suboptimal exposure of the IA [9].

We therefore decided to investigate these details as risk factors for lower success. In the thoracoscopic population, patients who underwent thymectomy, pericardiotomy and pledgeted sutures had a success rate similar to the sternotomy approach, while those with simple sutures, intact pericardium and no thymectomy had a lower success rate. Considering these results, we believe that the superior outcomes reported by Sutton et al. [27] may be due to technical details rather than just a learning curve effect.

Therefore, the success of aortopexy seems to be dependent on surgical technical aspects rather than the approach used. In light of similar outcomes, it may be reasonable to consider the thoracoscopic approach preferable, considering the general advantages of minimally invasive surgery [45,46]. However, this cannot be recommended since not enough data specific to thoracoscopic aortopexy is currently available.

One of the main limitations of this review is the retrospective nature of all included papers. Given the rarity of the condition, it is unlikely that any RCT will be available soon. Moreover, the different classification systems and the intrinsic subjectivity in diagnosis make comparing different experiences difficult.

Given these concerns, the overall confidence in the results of these studies is limited. The lack of randomized controlled trials and prospective comparative studies means that findings must be interpreted with caution. The available evidence suggests that aortopexy is an effective intervention for severe tracheomalacia, but the true magnitude of its benefit and long-term durability remain uncertain due to the high risk of bias in the current literature.

Nonetheless, we believe that a systematic review of all papers published in the last 10 years analyzing over 400 patients provides a comprehensive overview and interesting points toward the optimization of this controversial surgery.

Other surgical options are available to treat TM and seem to be associated with good results [44].

Posterior tracheopexy is, at the moment, the most popular surgical alternative to anterior aortopexy for the treatment of tracheomalacia [47]. Some centers have switched their practice, adopting posterior tracheopexy as the first option [9,34].

This procedure consists of dissecting the common wall between the esophagus and the trachea, exposing the posterior tracheal wall. After moving the esophagus laterally, the posterior tracheal wall is sutured to the longitudinal spinal ligament, preventing the inward movement of the membranous trachea, thus reducing the expiratory collapse [47].

Posterior tracheopexy has been reported to be associated with comparable outcomes as aortopexy [24]. This technique has the theoretical advantage of directly assessing the tracheal collapse by fixing the pars membranacea to the anterior spine ligament after moving the esophagus aside. Moreover, it has the advantage of avoiding operating on big vessels, thus decreasing the risk of severe intra-operative hemorrhage. On the other hand, posterior tracheopexy has the disadvantage of re-operating on the posterior mediastinum, which in the majority of cases would have significant scar tissue, the result of an esophageal atresia repair. Furthermore, the possibility of worsening the esophageal dysmotility proper of an esophageal atresia may make some surgeons favor the anterior approach. This latter concern, however, seems mitigated by the recent study published by Torre et al. [48].

Posterior tracheopexy has been recently proposed at the time of primary repair of esophageal atresia [49]. A significant reduction in respiratory morbidity has been described in the first year of life compared to patients who did not undergo concomitant posterior tracheopexy [50]. Posterior tracheopexy has been gaining consensus among pediatric surgeons. However, comparative studies are not available at the moment. Further studies may contribute to defining the role of this surgical option in the treatment of tracheomalacia. An international multicenter randomized controlled trial (PORTRAIT trial) has been recently proposed and will hopefully provide robust evidence on the role of primary PT [51].

Stenting of the trachea is fascinating and potentially the least invasive option for the treatment of TM. Unfortunately, positioning a stent in the presence of a vascular extrinsic compression is contraindicated due to the risk of tracheal wall ischemia, necrosis and perforation [52]. Moreover, airway stents are associated with high complication rates, such as dislocation, granulation tissue formation and mucous plug [53]. More recently, absorbable stents have shown a decreased amount of complication, but they offer a transient benefit [54]. They are sometimes used as a bridge to surgical correction to guarantee airway patency or, in selected cases, to treat residual malacia after surgery [55,56]. At the moment, no consensus exists on the use of airway stents for pediatric tracheomalacia [3].

Unfortunately, no comparative studies among different treatment options are available to date, making it impossible to determine the superiority of any therapeutic option.

Future research should focus on prospective, multicenter studies with standardized outcome assessments, longer follow-up periods and improved control of confounding factors to strengthen the evidence base for aortopexy or other treatments for tracheomalacia management.

Further studies are desirable to achieve a validated and standardized classification and outcome reporting system in order to improve the comparability of different centers.

## 5. Conclusions

Aortopexy may be an effective surgical option for the treatment of tracheomalacia, but no strong recommendation can be given based on the current literature. There seems to be a general consensus in recommending this surgery for life-threatening symptoms associated with tracheomalacia, particularly with tracheal collapse greater than 70%. However, indications for non-life-threatening symptoms are not well defined. Thymectomy, pericardiotomy and pledgeted sutures appear to ensure a better success rate. When these steps are followed, thoracoscopy seems a valid approach. However, the superiority of the thoracoscopic approach is yet to be proven.

Prospective randomized comparative studies are desirable to better define TM severity and classification, surgical indications and to compare different surgical options.

## Figures and Tables

**Figure 1 jcm-14-01367-f001:**
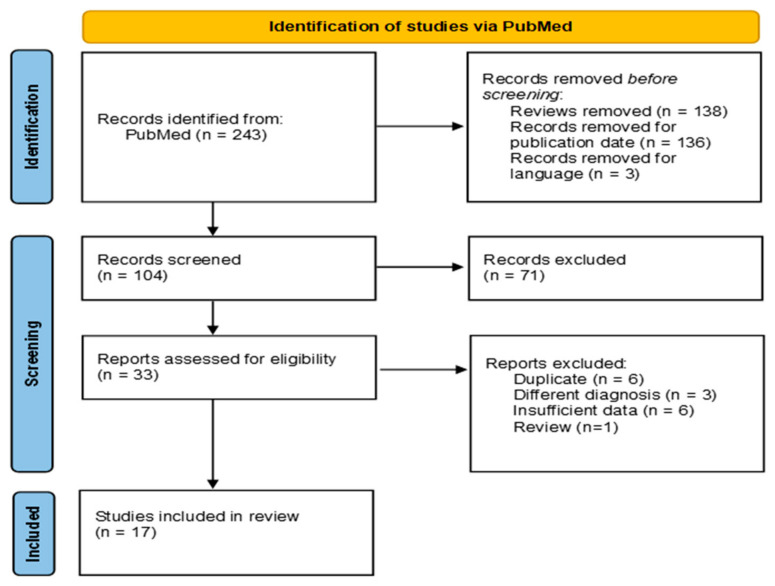
Study selection.

**Figure 2 jcm-14-01367-f002:**
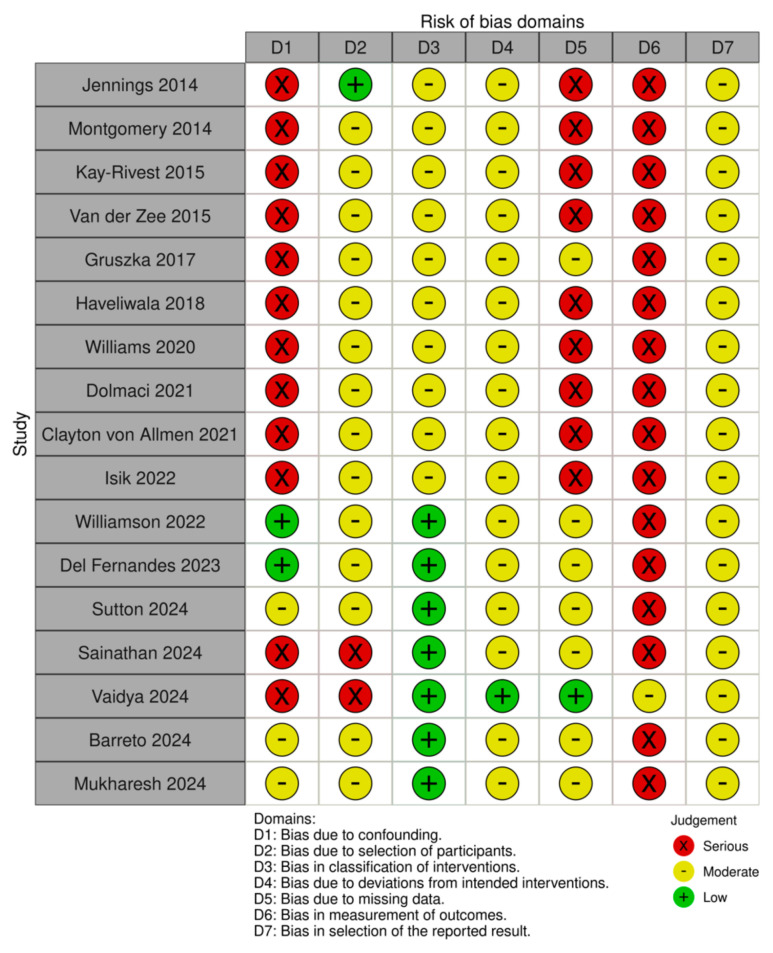
Visual representation of the risk of bias assessment using the ROBINS-1 tool (ref. [9,10,14,15,16,17,18,19,20,21,22,24,25,26,27,28]).

**Figure 3 jcm-14-01367-f003:**
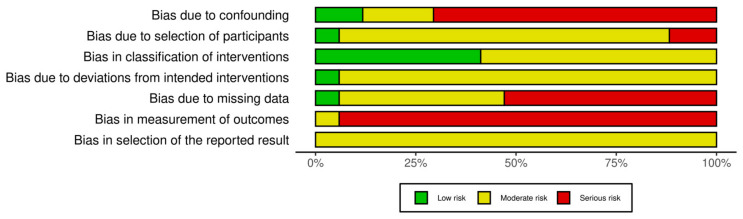
Visual representation of distribution of risk of bias judgment within each bias domain.

**Table 1 jcm-14-01367-t001:** Summary of the findings (Abbreviations: ST: Sternotomy; TT: Thoracotomy; TS: Thoracoscopy; CV: Cervical; NM: Not mentioned; PNX: Pneumothorax).

Author	Period	N°	Age Months (Range)	Approach	Thymectomy	Pericardiotomy	Pledgets	Complications	Redo	Success	Follow-Up
Jennings et al. *J. Pediatr. Surg.*, *2014* [9]	1997–2012	41	7.5 (1–136)	20 ST 13 TT 8 TS	Partial, only open cases	20/20 ST 13/13 TT 5/8 TS	20/20 ST	1 vocal cord palsy	3 TS	20/20 ST 10/13 TT 5/8 TS	10 m–14 yr
Montgomery et al. *Eur. J. Pediatr. Surg.*, *2014* [14]	1993–2012	30	6.8 (0.5–31)	20 TT 10 ST	NM	NM	NM	2 PNX 1 vocal cord palsy	0	22/30 3/30 partial relief	25.5 (1 m–12 yr)
Kay-Rivest et al. *Dis. Esophagus.*, *2015* [15]	1989–2010	6	2.1 (0.6–3)	6 TT	6 total	NM	NM	1 bleeding 1 phrenic nerve palsy	0	6/6	2 yr
Van der Zee et al. *World J. Surg.*, *2015* [10]	2002–2012	16	5 (0.5–12)	16 TS	No	No	No	0	5	10/16	6 m–10 yr
Gruszka et al. *Interact. Cardiovasc. Thorac. Surg.*, *2017* [16]	1994–2012	53	13 (1–120)	29 ST 21 TT	Partial or total	No	53/53	1 pleural effusion 3 phrenic nerve palsy	0	51/53	4.9 yr (0.3–14.9)
Haveliwala et al. *J. Pediatr. Surg.*, *2018* [17]	2016–2018	22	5 (0.5–60)	22 CV	Partial or total	NM	22/22	1 PNX	0	16	6 wks
Williams et al. *J. Laryngol. Otol.*, *2020* [18]	2007–2017	25	9.4 (0.5–35)	23 ST 2 TT	Total	NM	NM	4 tracheostomy 3 respiratory infections 2 vocal cord palsy 2 wound infection	1	20	5.2 yr (1.2–8.5)
Dolmaci et al. *Interact. Cardiovasc. Thorac. Surg.*, *2021* [19]	2010–2020	24	9 (2–117)	24 ST	20/24 partial or total	24/24	Straps	1 PNX 1 pericardial effusion	0	22	25.5 m (18–34)
Clayton von Allmen et al. *Int. J. Pediatr. Otorhinolaryngol.*, *2021* [28]	2011–2021	10	102.8 (3–192)	10 CV	Partial or total	NM	10/10	1 seroma 1 reintubation	1	6	NM
Isik et al. *Pediatr. Surg. Int.*, *2022* [20]	2018–2021	15	16.3 (1–31)	15 ST	Partial	NM	15/15	NM	NM	NM	NM
Williamson et al. *J Pediatr Surg2022* [21]	2010–2010	10	2.2 (0–6)	NM	NM	NM	NM	NM	NM	NM	NM
Del Fernandes et al. *J. Pediatr. Surg.*, *2023* [22]	2000–2018	7	5 (1–15)	NM	NM	NM	NM	1 vocal cord palsy	0	7	NM
Sutton et al. *J. Pediatr. Surg.*, *2024* [27]	2006–2021	169	ST and TT: 7.7 (0–233) TS: 5.2 (2–27)	135 ST 12 TT 22 TS	NM ST and TT 22/22 TS partial	135/135 ST 12/12 TT 22/22 TS	135/135 ST 12/12 TT 22/22 TS	42 (PNX most common)	9 ST 2 TS	NM ST and TT 19/22 TS	8.6 yr (1–20 yr)
Sainathan et al. *Transl. Pediatr.*, *2024* [26]	2017–2020	9	9.6 (2–24)	9 ST	9/9 total	9/9	9/9	NM	0	7/9	6 m
Vaidya et al. *Innovations (Phila.)*, *2024* [29]	2024	1	3	1 ST	No	1/1	1/1	1 Dressler syndrome	0	1	6 m
Barreto et al. *J. Pediatr. (Rio J.)*, *2024* [24]	2003–2023	15	NM	Open approach	Partial or total	NM	15/15	NM	0	11	NM
Mukharesh et al. *Pediatr. Pulmonol.*, *2024* [25]	2013–2020	20	NM	20 ST	NM	NM	NM	NM	NM	NM	NM

## Data Availability

The raw data supporting the conclusions of this article will be made available by the authors on request.

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
