# Peer review of "Aortopexy for Tracheomalacia in Children: A Systematic Review and Meta-Analysis"

_jcm, 2025, doi:10.3390/jcm14041367_

Round 1

Reviewer 1 Report

Comments and Suggestions for Authors

The manuscript (AORTOPEXY FOR TRACHEOMALACIA IN CHILDREN: A COMPREHENSIVE REVIEW) provides a review for a clinically relevant topic.

I have the following comments:

The topic is not new and the conclusion does not add to the current knowledge

Title should contain systematic review

The introduction section is very deficient. What knowledge gap is currently present to justify this review?

Definitions of vague terms are required, such as severity of tracheomalacia.

Reasons for excluding some studies are needed

Quality assessment is required

Statistical analysis makes this a meta-analysis, not a systematic review

Longitudinal data is crucial for this topic

Did you consider comparisons with other techniques?

Discuss reasons for variability of the results between different centers

Add primsa check list

Add Prisma flowchart

What are the recommendations from this review?

Author Response

Comments and Suggestions for Authors

The manuscript (AORTOPEXY FOR TRACHEOMALACIA IN CHILDREN: A COMPREHENSIVE REVIEW) provides a review for a clinically relevant topic.

I have the following comments:

The topic is not new and the conclusion does not add to the current knowledge

We are aware the topic is not new. We intended to update the last review published on this topic which is from 2012. Compared to that review, thoracoscopic approach has largely developed and data presented in our review are new compared to the previous review. We believe that our overview of all published data provides interesting information.

Title should contain systematic review

The term “systematic review” was abandoned on request of the academic editor. We are glad to accept your suggestion and change the title back to systematic review.

The introduction section is very deficient. What knowledge gap is currently present to justify this review?

Thank for your suggestion. We developed the introduction section stating the gap we intended to fill.

Definitions of vague terms are required, such as severity of tracheomalacia.

We specified the definition of tracheomalacia severity

Reasons for excluding some studies are needed

Reasons for study exclusion are detailed in the PRISMA flowchart in Table 1. We apologize that table 1 was not uploaded

Quality assessment is required

Quality assessment has been made and provided

Statistical analysis makes this a meta-analysis, not a systematic review

We changed the definition of our study accordingly

Longitudinal data is crucial for this topic

We agree on this. Unfortunately, no longitudinal data has been previously published hence are not reported in our review. We included this aspect on the limitations.

Did you consider comparisons with other techniques?

This is the focus on a following study we are currently conducting

Discuss reasons for variability of the results between different centers

We added a brief discussion

Add primsa check list

Prisma check list has been added

Add Prisma flowchart

Prisma flowchart has been added

Reviewer 2 Report

Comments and Suggestions for Authors

Zanini et al reported their manuscript titled "Aortopexy for Tracheomalacia in Children: A Comprehensive Review". I have the following comments:

General Comments:

  1. Strengths:
    • The manuscript provides a comprehensive review of aortopexy for tracheomalacia (TM) in children, covering a wide range of surgical approaches, outcomes, and technical details.
    • The systematic review follows PRISMA guidelines, which enhances the methodological rigor.
    • The inclusion of 17 studies with 473 patients offers a robust dataset for analysis.
    • The discussion on the controversies surrounding aortopexy, including surgical indications, approaches, and success rates, is thorough and well-structured.
  2. Weaknesses:
    • The retrospective nature of the included studies limits the ability to draw definitive conclusions, particularly regarding causality and long-term outcomes.
    • The lack of standardized definitions for TM severity and surgical success across studies makes it difficult to compare outcomes.
    • The review does not sufficiently address the potential biases in the included studies, such as selection bias or reporting bias.
    • The discussion on alternative surgical techniques (e.g., posterior tracheopexy) is limited, and no comparative analysis is provided.

Specific Comments:

1. Introduction:

  • The introduction provides a good overview of the history and current controversies surrounding aortopexy. However, it could benefit from a clearer statement of the review's objectives and research questions. Please consider adding a specific aim or hypothesis to guide the reader, such as "This review aims to evaluate the outcomes of aortopexy in children with tracheomalacia and identify factors associated with procedural success."

2. Materials and Methods:

  • The PRISMA guidelines were followed, which is a strength. Please clarify if grey literature or conference proceedings were included. Please try to add PRISMA flowchart (optional request)

3. Results:

  • The results section is well-organized, but some data are missing or inconsistently reported across studies (e.g., weight at surgery, follow-up duration). Please acknowledge these limitations in the discussion and consider performing a sensitivity analysis to assess the impact of missing data.
  • The success rate of aortopexy is reported as 84%, but the definition of "success" varies across studies. Please provide a standardized definition of success (e.g., complete resolution of symptoms, improvement in quality of life) and discuss how this variability may affect the results.

4. Discussion:

  • The discussion is comprehensive but could benefit from a more critical analysis of the limitations of the included studies. Please discuss the potential for publication bias, as negative outcomes may be underreported. Additionally, consider the impact of heterogeneity in surgical techniques and patient populations on the results.
  • The discussion on thoracoscopic vs. open approaches is insightful, but the conclusion that thoracoscopy is preferable is not fully supported by the data. Please emphasize that the superiority of thoracoscopy is not yet proven and that more data are needed before recommending it as the preferred approach.
  • The discussion on alternative surgical techniques (e.g., posterior tracheopexy) is limited. Please expand this section to include a more detailed comparison of outcomes, advantages, and disadvantages of alternative techniques.

5. Conclusion:

  • The conclusion is well-written but could be more specific about the need for future research. Please highlight the need for prospective studies, standardized definitions of TM severity, and comparative studies of different surgical techniques.

Minor Comments:

1.       Tables and Figures: Table 1 and Table 2 are mentioned in the text but are not provided in the manuscript. Please ensure that all tables and figures are included and referenced correctly.

  1. References: The references are comprehensive, but please consider including more recent studies to provide an up-to-date perspective on aortopexy and tracheomalacia.
  2. Language and Clarity: Please simplify complex sentences and avoid redundancy, particularly in the discussion section.
Comments on the Quality of English Language

 Please simplify complex sentences and avoid redundancy, particularly in the discussion section.

Author Response

Comments and Suggestions for Authors

Zanini et al reported their manuscript titled "Aortopexy for Tracheomalacia in Children: A Comprehensive Review". I have the following comments:

General Comments:

Strengths:

The manuscript provides a comprehensive review of aortopexy for tracheomalacia (TM) in children, covering a wide range of surgical approaches, outcomes, and technical details.

The systematic review follows PRISMA guidelines, which enhances the methodological rigor.

The inclusion of 17 studies with 473 patients offers a robust dataset for analysis.

The discussion on the controversies surrounding aortopexy, including surgical indications, approaches, and success rates, is thorough and well-structured.

Weaknesses:

The retrospective nature of the included studies limits the ability to draw definitive conclusions, particularly regarding causality and long-term outcomes.

We agree. Unfortunately, no prospective study has been published to date. We hope some will be soon available.

The lack of standardized definitions for TM severity and surgical success across studies makes it difficult to compare outcomes.

We agree. This is one of the main issues in the comparison between different Centre’s experience on Tracheomalacia. We are currently working on the development and validation of an objective tool to define TM severity.

The review does not sufficiently address the potential biases in the included studies, such as selection bias or reporting bias.

We added the bias assessment and commented on the potential biases.

The discussion on alternative surgical techniques (e.g., posterior tracheopexy) is limited, and no comparative analysis is provided.

Posterior tracheopexy has been recently introduced and at the moment not enough data are available for a a comparative analysis.

Specific Comments:

  1. Introduction:

The introduction provides a good overview of the history and current controversies surrounding aortopexy. However, it could benefit from a clearer statement of the review's objectives and research questions. Please consider adding a specific aim or hypothesis to guide the reader, such as "This review aims to evaluate the outcomes of aortopexy in children with tracheomalacia and identify factors associated with procedural success."

Thanks for the suggestion. We added the suggested statement.

  1. Materials and Methods:

The PRISMA guidelines were followed, which is a strength. Please clarify if grey literature or conference proceedings were included. Please try to add PRISMA flowchart (optional request)

We stated that grey literature and conference were not searched. We apologize for having not uploaded the PRISMA flowchart, it has been uploaded now.

  1. Results:

The results section is well-organized, but some data are missing or inconsistently reported across studies (e.g., weight at surgery, follow-up duration). Please acknowledge these limitations in the discussion and consider performing a sensitivity analysis to assess the impact of missing data.

We acknowledged this limitation in the discussion section

The success rate of aortopexy is reported as 84%, but the definition of "success" varies across studies. Please provide a standardized definition of success (e.g., complete resolution of symptoms, improvement in quality of life) and discuss how this variability may affect the results.

We expanded our definition of “success” in the methods section

  1. Discussion:

The discussion is comprehensive but could benefit from a more critical analysis of the limitations of the included studies. Please discuss the potential for publication bias, as negative outcomes may be underreported. Additionally, consider the impact of heterogeneity in surgical techniques and patient populations on the results.

The discussion on thoracoscopic vs. open approaches is insightful, but the conclusion that thoracoscopy is preferable is not fully supported by the data. Please emphasize that the superiority of thoracoscopy is not yet proven and that more data are needed before recommending it as the preferred approach.

The discussion on alternative surgical techniques (e.g., posterior tracheopexy) is limited. Please expand this section to include a more detailed comparison of outcomes, advantages, and disadvantages of alternative techniques.

We expanded the discussion according to the suggestions

  1. Conclusion:

The conclusion is well-written but could be more specific about the need for future research. Please highlight the need for prospective studies, standardized definitions of TM severity, and comparative studies of different surgical techniques.

We provided the suggested considerations

Minor Comments:

  1. Tables and Figures: Table 1 and Table 2 are mentioned in the text but are not provided in the manuscript. Please ensure that all tables and figures are included and referenced correctly.

We apologize for the inconvenience. Table and figure have been re-submitted

References: The references are comprehensive, but please consider including more recent studies to provide an up-to-date perspective on aortopexy and tracheomalacia.

We included the most recent and relevant references

Language and Clarity: Please simplify complex sentences and avoid redundancy, particularly in the discussion section.

We edited the paper accordingly.

Reviewer 3 Report

Comments and Suggestions for Authors

General Comments:

  • The review is a comprehensive synthesis of current evidence on aortopexy in pediatric tracheomalacia, adhering to PRISMA guidelines. The topic is relevant, given the controversies surrounding surgical indications and approaches.
  • The manuscript is well-structured, with clearly defined sections and sufficient methodological details. However, it has some areas that need improvement to enhance clarity and scientific rigor.

Major Comments:

  1. Definition of Tracheomalacia Severity:

    • While the manuscript discusses the lack of uniform classification for tracheomalacia severity, it does not provide sufficient analysis of the implications of this heterogeneity on clinical decision-making or outcomes. Consider elaborating on how inconsistent definitions impact surgical outcomes and recommendations.
  2. Data Interpretation:

    • The discussion on success rates between open and thoracoscopic approaches is insightful. However, the manuscript should emphasize the potential bias from the retrospective nature of included studies and the learning curve effect in minimally invasive surgery.
    • The lack of analysis on predictors of surgical outcomes and complications is a limitation. Further stratification (e.g., by age, underlying conditions) could provide more actionable insights.
  3. Incomplete Data Reporting:

    • Many included studies do not provide complete data on thymectomy, pericardiotomy, or suture techniques. While this limitation is acknowledged, consider suggesting specific guidelines for future reporting in this area.
  •  
  • Figures and Tables:
    • The addition of a flowchart for PRISMA and detailed tables summarizing outcomes (success rates, complications) by surgical approach would improve the readability and utility of the manuscript.
  • References:
    • Some references are outdated or incomplete. Ensure all references are correctly formatted and include the most recent studies where possible.

Specific Recommendations:

  1. Expand the discussion on posterior tracheopexy, as it was only briefly mentioned.
  2. Clarify the criteria used to define "success" in different studies and suggest standardization.
  3. Discuss ethical considerations, especially given the lack of prospective trials in this field.
Comments on the Quality of English Language
  • Language:
    • There are typographical errors in several sections (e.g., "matherials" instead of "materials" and "aempt" instead of "attempt"). These should be corrected.
  • Sentence Structure:

    • Some sentences are overly long or awkwardly constructed, which can make them difficult to follow. For example, rephrasing complex sentences into shorter, more concise statements would improve clarity.
  • Consistency in Terminology:

    • Ensure that technical terms and phrases, such as "tracheomalacia severity" and "success rate," are consistently used throughout the manuscript.
  • Grammar and Word Choice:

    • A few grammatical errors and awkward phrases should be addressed. For instance, some verbs and prepositions could be replaced with more appropriate alternatives to improve fluency.
  • Style:

    • Adopting a more formal and academic tone in certain sections, especially in the discussion, would enhance the overall presentation.

Author Response

Comments and Suggestions for Authors

General Comments:

The review is a comprehensive synthesis of current evidence on aortopexy in pediatric tracheomalacia, adhering to PRISMA guidelines. The topic is relevant, given the controversies surrounding surgical indications and approaches.

The manuscript is well-structured, with clearly defined sections and sufficient methodological details. However, it has some areas that need improvement to enhance clarity and scientific rigor.

Major Comments:

Definition of Tracheomalacia Severity:

While the manuscript discusses the lack of uniform classification for tracheomalacia severity, it does not provide sufficient analysis of the implications of this heterogeneity on clinical decision-making or outcomes. Consider elaborating on how inconsistent definitions impact surgical outcomes and recommendations.

We added a comment on this limitation.

Data Interpretation:

The discussion on success rates between open and thoracoscopic approaches is insightful. However, the manuscript should emphasize the potential bias from the retrospective nature of included studies and the learning curve effect in minimally invasive surgery.

We expanded the discussion regarding the impact of potential biases

The lack of analysis on predictors of surgical outcomes and complications is a limitation. Further stratification (e.g., by age, underlying conditions) could provide more actionable insights.

We added an analysis on predictors factor.

Incomplete Data Reporting:

Many included studies do not provide complete data on thymectomy, pericardiotomy, or suture techniques. While this limitation is acknowledged, consider suggesting specific guidelines for future reporting in this area.

 Thank you for your suggestion. We added this recommendation.

Figures and Tables:

The addition of a flowchart for PRISMA and detailed tables summarizing outcomes (success rates, complications) by surgical approach would improve the readability and utility of the manuscript.

PRISMA flowchart and table have been added

References:

Some references are outdated or incomplete. Ensure all references are correctly formatted and include the most recent studies where possible.

References have been updated

Specific Recommendations:

Expand the discussion on posterior tracheopexy, as it was only briefly mentioned.

Discussion on posterior tracheopexy has been expanded

Clarify the criteria used to define "success" in different studies and suggest standardization.

We highlighted the definition of “success” and suggested standardization.

Discuss ethical considerations, especially given the lack of prospective trials in this field.

Comments on the Quality of English Language

Language:

There are typographical errors in several sections (e.g., "matherials" instead of "materials" and "aempt" instead of "attempt"). These should be corrected.

We corrected the typographical errors.

Sentence Structure:

Some sentences are overly long or awkwardly constructed, which can make them difficult to follow. For example, rephrasing complex sentences into shorter, more concise statements would improve clarity.

We simplified the long sentences.

Consistency in Terminology:

Ensure that technical terms and phrases, such as "tracheomalacia severity" and "success rate," are consistently used throughout the manuscript.

Thank you for you comment. We verified the terminology consistency

Grammar and Word Choice:

A few grammatical errors and awkward phrases should be addressed. For instance, some verbs and prepositions could be replaced with more appropriate alternatives to improve fluency.

We reviewed and corrected the paper

Style:

Adopting a more formal and academic tone in certain sections, especially in the discussion, would enhance the overall presentation.

We changed the paper trying to adopt a more formal tone.

Round 2

Reviewer 1 Report

Comments and Suggestions for Authors

Thank you for modifying the manuscript in response to the previous comments. The manuscript has improved markedly. however, there is no change made to the introduction. The knowledge gap and the need for such research had not been addressed. I recommend expanding the introduction section and reinforce it with studies from the literature.

Author Response

Thank you for modifying the manuscript in response to the previous comments. The manuscript has improved markedly. however, there is no change made to the introduction. The knowledge gap and the need for such research had not been addressed. I recommend expanding the introduction section and reinforce it with studies from the literature.

Thanks for your rating and suggestions. We have made changes to the introduction

Reviewer 2 Report

Comments and Suggestions for Authors

The authors have addressed my prior comments.

Author Response

The authors have addressed my prior comments.

response: thank you for your rating